# A Critical Regulation of Th17 Cell Responses and Autoimmune Neuro-Inflammation by Ginsenoside Rg3

**DOI:** 10.3390/biom10010122

**Published:** 2020-01-10

**Authors:** Young-Jun Park, Minkyoung Cho, Garam Choi, Hyeongjin Na, Yeonseok Chung

**Affiliations:** 1Laboratory of Immune Regulation, Research Institute of Pharmaceutical Sciences, College of Pharmacy, Seoul National University, Seoul 08826, Korea; flavah00@gmail.com (Y.-J.P.); grhappy@snu.ac.kr (G.C.); hyeongjinna@snu.ac.kr (H.N.); 2Brain Korea 21 Program, College of Pharmacy, Seoul National University, Seoul 08826, Korea

**Keywords:** Rg3, Th17, RORγt, EAE

## Abstract

Among diverse helper T-cell subsets, Th17 cells appear to be pathogenic in diverse autoimmune diseases, and thus, targeting Th17 cells could be beneficial for the treatment of the diseases in humans. Ginsenoside Rg3 is one of the most potent components in Korean Red Ginseng (KRG; *Panax ginseng* Meyer) in ameliorating inflammatory responses. However, the role of Rg3 in Th17 cells and Th17-mediated autoimmunity is unclear. We found that Rg3 significantly inhibited the differentiation of Th17 cells from naïve precursors in a dendritic cell (DC)–T co-culture system. While Rg3 minimally affected the secretion of IL-6, TNFα, and IL-12p40 from DCs, it significantly hampered the expression of IL-17A and RORγt in T cells in a T-cell-intrinsic manner. Moreover, Rg3 alleviated the onset and severity of experimental autoimmune encephalomyelitis (EAE), induced by transferring myelin oligodendrocyte glycoprotein (MOG)-reactive T cells. Our findings demonstrate that Rg3 inhibited Th17 differentiation and Th17-mediated neuro-inflammation, suggesting Rg3 as a potential candidate for resolving Th17-related autoimmune diseases.

## 1. Introduction

Due to their immunomodulatory functions, several herbal medicines have been used for the treatment of immunological disorders [1]. Among them, Korean Red Ginseng (KRG) has been traditionally prescribed for various diseases, although the exact mechanisms by which KRG mitigates the severity of diseases remain unclear [2]. Recent advances in understanding the pharmacological components of KRG shed light on identifying a variety of bioactive ingredients, such as ginsenosides, polysaccharides, phytosterols, peptides, polyacetylenic alcohols, and fatty acids, that prevent and eradicate diabetes, tumors, ulcers, aging, and depression [3,4,5,6].

It has been suggested that ginsenosides are among the most potent biologically active constituents of KRG [7]. More than 100 types of ginsenosides are present, stratified by the differential polarity attributed to their chemical backbones [8]. Interestingly, numerous studies have reported that anti-inflammatory effects are common features of a myriad of ginsenosides [2,9]. For example, enhanced Rg3, one of the major ginsenosides present in KRG, inhibits IFNγ-producing helper CD4^+^ T cells (Th1) that prevalently expand in the conditions of excessive inflammation [10]. Rg3 also promotes immunosuppressive M2 macrophage polarization, which can resolve self-destructive immune responses [11]. Moreover, increasing numbers of recent studies demonstrate that ginsenosides can ameliorate the disease severity in animal models of multiple sclerosis, Crohn’s disease, and rheumatoid arthritis, suggesting that ginsenosides modulate autoimmune responses in vivo [1].

Autoimmune diseases are caused by unnecessarily aggressive autoreactive CD4^+^ T-cell responses that attack self-tissue and/or self-antigen. Among diverse CD4^+^ T-cell subsets, IL-17A-producing CD4^+^ T cells (Th17) provide protection against extracellular pathogens and fungi, but also exert detrimental roles in mediating tissue inflammation in autoimmune diseases [12,13]. Indeed, antagonistic antibodies to IL-17A or IL-17RA have shown a clinical benefit in patients with psoriasis [14]. The differentiation of Th17 cells is induced by IL-6 and TGFβ, which activate STAT3 and subsequently induce RORγt [15,16,17]. Recently, two independent studies have demonstrated that KRG can inhibit Th17 cell differentiation by hampering STAT3 activation, leading to the amelioration of collagen-induced arthritis and alloantigen-induced inflammation [18,19]. Our previous study also suggested a role of Rg3 in T-cell differentiation [10], indicating a possible role of ginsenosides in T-cell-mediated immune disorders.

In the present study, we aimed to investigate whether ginsenoside Rg3 modulated Th17 differentiation and Th17-mediated experimental autoimmune encephalomyelitis (EAE), in an animal model for human multiple sclerosis. By using multiple in vitro T-cell culture systems, we found that Rg3 inhibited Th17 cell differentiation in a T-cell-intrinsic manner. Moreover, Rg3 alleviated the incidence and severity of EAE in vivo.

## 2. Materials and Methods

### 2.1. Ethics Statement

All animal experiments were approved by the Institutional Animal Care and Use Committee of Seoul National University (IACUC protocol #: SNU-170120-1) and were conducted in accordance with guidelines of Seoul National University for the care and use of laboratory animals.

### 2.2. Mice and EAE Model

Six-week-old C57BL/6 female mice were purchased from Orient Bio (Gyeonggi, Korea). For T-cell-transfer EAE, mice were immunized with 300 μg myelin oligodendrocyte glycoprotein peptide (MOG_35–55_) peptide emulsified in complete freund’s adjuvant (CFA) with heat-inactivated *Mycobacterium tuberculosis*. Seven days after immunization, cells from draining lymph nodes were isolated and cultured with 20 μg/mL MOG_35–55_ and IL-23 (20 ng/mL) (PeproTech, Rocky Hill, NJ, USA). On day five post in vitro stimulation, cells were harvested and enriched with CD4^+^ T cells by using magnetic beads. The enriched CD4^+^ T cells (2 × 10^6^ cells/mouse) were transferred into mice followed by immunization with MOG_35–55_ in CFA (s.c.) and subsequent pertussis toxin (PT) (i.p.) injection, and the clinical severity for EAE pathology was monitored daily as previously mentioned [20].

### 2.3. Bone-Marrow-Derived Dendritic Cells (BMDCs) Generation

Bone marrow cells obtained from C57BL/6 mice were cultured in PRMI1640 supplemented with 10% FBS, 55 μM 2-mercaptoethanol, penicillin/streptomycin at 1 × 10^6^ cells/mL (All from Gibco, Grand Island, NY, USA), and recombinant mouse GM-CSF (10 ng/mL) (PeproTech, Rocky Hill, NJ, USA). Twenty-four hours later, floating cells were transferred into new plates. Fresh medium was added every other day. On day seven, cells were recovered and CD11c^+^ cells were isolated using magnetic beads, and the sorted CD11c^+^ DCs were used in further studies.

### 2.4. In Vitro Th17 Cell Differentiation

Naïve CD4^+^ T cells (CD44^low^CD62L^high^CD25^−^) were isolated from wild-type mice using a cell sorter (BD BioScience, San Jose, CA, USA). For DC-mediated Th17 cell differentiation, BMDCs were co-cultured with naive CD4^+^ T cells in the presence of anti-CD3ε (0.3 μg/mL) antibody (145-2C11, BioXcell, NH, USA), TGFβ (1.5 ng/mL) (PeproTech, Rocky Hill, NJ, USA), and LPS (100 ng/mL) (Sigma, Seoul, Korea) for 96 h. For DC-free Th17 cell differentiation, anti-CD3ε (1 μg/mL) and anti-CD28 (1 μg/mL) (37.51, BioXcell, West Lebanon, NH, USA) were pre-coated in a 96-well flat-bottom plate overnight at 4 °C. After washing the plate with cold PBS three times, 1 × 10^5^ naïve CD4^+^ T cells were stimulated with IL-6 (10 ng/mL) and TGFβ (1.5 ng/mL) (PeproTech, Rocky Hill, NJ, USA) for 96 h. For Rg3 (LKT Labs, Saint Paul, MN, USA; dissolved in DMSO) treatment, various concentrations (9.37, 18.75, and 37.5 μg/mL) were added at the beginning of in vitro culture.

### 2.5. ELISA

IL-6, 12p40, IL-17A, and TNFα in the culture supernatants of LPS-stimulated BMDCs or T cells stimulated under Th17-skewing conditions were quantified by ELISA according to the manufacturer’s instructions (eBioscience, San Diego, CA, USA).

### 2.6. Real-Time RT-PCR

Total RNA from cells was isolated by TRIzol reagent (Ambion, Austin, TX, USA) and cDNA was synthesized with a cDNA Synthesis kit (Thermo Fisher Scientific Inc., Waltham, MA, USA). Relative gene expression levels were evaluated using SYBR Green (Bio-Rad, Philadelphia, PA, USA) on ABI 7500 Fast Real-Time PCR Systems (Applied Biosystems, Singapore). Target genes were normalized to the *Hprt* level in each sample. Primer sets for genes were synthesized at Cosmogenetech (Seoul, Korea): *Il6* (sense, 5′- TCG GAG GCT TAA TTA CAC ATG TTC T -3′, antisense, 5′- GCA TCA TCG TTG TTC ATA CAA TCA -3′), *Il12b* (sense, 5′- AAA CCA GAC CCG CCC AAG AAC -3′, antisense, 5′- AAA AAG CCA ACC AAG CAG AAG ACA G -3′), *Il17a* (sense, 5′- CTC CAG AAG GCC CTC AGA CTA C -3′, antisense, 5′- GGG TCT TCA TTG CGG TGG -3′), *Il22* (sense, 5′- CAT GCA GGA GGT GGT ACC TT -3′, antisense, 5′- CAG ACG CAA GCA TTT CTC AG -3′), *Rorc* (sense, 5′-CCG CTG AGA GGG CTT CAC -3′, antisense, 5′- TGC AGG AGT AGG CCA CAT TAC A -3′),*Tnfa* (sense, 5′- ATG AGA AGT TCC CAA ATG GCC -3′, antisense, 5′- TCC ACT TGG TGG TTC GCT ACG -3′), *Hprt* (sense, 5′- GGT TAA GCA GTA CAG CCC CAA AAT -3′, antisense, 5′- ATA GGC ACA TAG TGC AAA TCA AAA GTC -3′).

### 2.7. Flow Cytometry Analysis

For intracellular cytokine staining, cells were incubated for 3 h with 100 ng/mL of PMA and 1 μM of ionomycin (all from Sigma-Aldrich, Saint Louis, MO, USA), brefeldin A, and monensin (all from eBioscience, San Diego, CA, USA). After washing cells with cold PBS containing 1.5% FBS, cells were stained with APC-Cy7-conjugated anti-CD45.2 mAb and PE/Cy7-conjugated anti-CD4 mAb (eBioscience, San Diego, CA, USA) for surface staining. Cells were then washed and stained with PerCp-Cy5.5-conjugated anti-IFNγ mAb, APC-conjugated anti-IL-17 mAb (all from BioLegend, San Diego, CA, USA) and PE-conjugated anti-RORγt mAb (eBioscience, San Diego, CA, USA) after incubation with fixation/permeabilization buffer (eBioscience, San Diego, CA, USA) for 30 min at 4 °C. Cells were analyzed by LSR III flow cytometer (BD Bioscience, San Jose, CA, USA). Data were analyzed with FlowJo (TreeStar, Ashland, OR, USA).

### 2.8. Statistical Analysis

All experiments were performed more than three times. Statistical analysis was conducted with mean ± SEM by unpaired two-tailed Student’s *t*-test with GraphPad Prism 5.0 (GraphPad Software Inc., San Diego, CA, USA).

## 3. Results

### 3.1. Rg3 Minimally Affects the Production of Th17-Promoting Cytokines from DCs

Our previous study showed that KRG extract and enhanced Rg3 suppressed pro-inflammatory cytokines production by LPS-stimulated DCs [10]. Moreover, KRG extract has been shown to inhibit Th17-cell differentiation in the presence of cyclosporine in vitro [18]. Hence, as a first step to determine the potential role of Rg3 in Th17 cell differentiation, we asked if Rg3 impacts the production of Th17-promoting cytokines, including IL-6, TNFα, and IL-12/IL-23p40 from DCs upon LPS treatment [21]. We generated bone-marrow derived DCs (BMDCs) and stimulated them with LPS for 12 h in the presence or absence of Rg3 (Figure 1A). DMSO was used as a vehicle control since Rg3 was dissolved in DMSO. We observed that, although Rg3 treatment slightly increased the transcript level of *Tnfa*, it had little effect on the level of TNFα production by the BMDCs (Figure 1B). Similarly, treatment with Rg3 had little effect on the transcript levels of *Il12b* and *Il6* as well as on the protein levels of IL-12p40 and IL-6 (Figure 1C,D). Thus, Rg3 treatment played only a minor role in the induction of pro-Th17 cytokines from DCs in this experimental setting.

### 3.2. Rg3 Inhibits Th17 Cell Differentiation in a T-Cell-Intrinsic Manner

To determine if Rg3 impacts Th17 cell differentiation, we employed a well-established DC-mediated Th17 cell differentiation system in which IL-17A-producing CD4^+^ T cells are induced by stimulating naïve CD4^+^ T cells with soluble anti-CD3, LPS, and TGFβ in the presence of BMDCs for four days [22] (Figure 2A). We observed about 10–13% of IL-17A^+^ cells by day four. However, upon treatment with Rg3, the frequency of IL-17A-producing Th17 cells was significantly decreased in a dose-dependent manner (Figure 2B). Accordingly, the production of IL-17A in the culture supernatant was significantly diminished by Rg3 treatment (Figure 2C). These results demonstrate that Rg3 can suppress the differentiation of Th17 cells from naïve precursors in vitro.

Since Rg3 treatment itself resulted in little change in the production of IL-6, IL-12/IL-23p40, and TNFα from BMDCs (Figure 1), we hypothesized that Rg3 might inhibit Th17 cell differentiation by acting on T cells rather than on DCs. To address the T-cell-modulatory effect of Rg3 during Th17 cell differentiation, we employed a DC-free Th17 cell differentiation system in which naïve CD4^+^ T cells were stimulated with plate-bound anti-CD3 and anti-CD28 antibodies in the presence of IL-6 and TGFβ for 4 days [22] (Figure 3A). Consistent with the observation in the DC-mediated Th17 cell differentiation system, we observed that the addition of Rg3 remarkably reduced the frequency of IL-17A-producing T cells and the production of IL-17A in the DC-free Th17 cell differentiation system (Figure 3B,C). Moreover, the transcript levels of Th17-cell-associated genes including *Il17a*, *Il21*, *Il22*, and *Rorc* in the T cells were all diminished by Rg3 (Figure 3D). These results together indicate that Rg3 inhibits Th17 cell differentiation by directly acting on CD4^+^ T cells rather than on DCs.

### 3.3. Rg3 Inhibits RORγt Expression in CD4^+^ T Cells during Th17 Cell Differentiation

Among pro-inflammatory cytokines secreted by DCs, IL-6 is critical for Th17 differentiation [23]. Our data demonstrated that Rg3 had little role in the production of IL-6 from DCs while inhibiting Th17 cell differentiation in the presence of IL-6 and TGFβ. Thus, we hypothesized that Rg3 modulates Th17 cell differentiation by regulating signals within the T cell itself. In the presence of TGFβ (1.5 ng/mL), IL-6 induces the expression of orphan nuclear receptor RORγt in a STAT3-dependent manner, which acts as the master transcription factor Th17 cell commitment [16]. To examine if Rg3 impacts the expression of RORγt in T cells, we stimulated naïve CD4^+^ T cells in the DC-free Th17-skewing condition with increasing doses of IL-6. Compared with 10 ng/mL of IL-6, 20 or 40 ng/mL of IL-6 induced an increase in IL-17A^+^ T cells up to ~40%. However, the same treatment resulted in ~10% of IL-17A^+^ T cells in the presence of Rg3 (Figure 4A). We observed a similar pattern in the frequency of IL-17A^+^ RORγt^+^ T cells (Figure 4B), suggesting a possible role of Rg3 in regulating RORγt expression in T cells. As shown in Figure 4C, stimulation with 10 ng/mL of IL-6 and TGFβ (1.5 ng/mL) induced the expression of RORγt in over 80% of CD4^+^ T cells. However, the addition of Rg3 significantly hampered the expression of RORγt in CD4^+^ T cells. Increasing IL-6 failed to restore the diminished RORγt in CD4^+^ T cells (Figure 4C). Thus, these results strongly suggest that Rg3 inhibits Th17 cell differentiation by hampering RORγt expression in CD4^+^ T cells.

### 3.4. Rg3 Attenuates the Reactivation of Autoreactive Th17 Cells and Experimental Autoimmune Encephalomyelitis Induced by MOG-Reactive CD4^+^ T-Cell Transfer

Inhibition of Th17 cell differentiation and RORγt expression by Rg3 treatment in vitro prompted us to examine if Rg3 also inhibits the reactivation of Th17 cells generated in vivo. Stimulation of the lymphoid cells from myelin oligodendrocyte glycoprotein peptide (MOG_35–55_)-immunized mice with MOG in the presence of IL-23 is known to reactivate and expand MOG-specific Th17 cells [24]. When we employed this experimental model (Figure 5A), we observed that ~40% of the CD4^+^ T cells expressed IL-17A and ~60% expressed IFNγ (Figure 5B), indicating that MOG-specific pathogenic Th17 cells were reactivated and expanded upon restimulation. By contrast, the addition of Rg3 during the restimulation step significantly diminished the frequency of IFNγ^−^IL-17A^+^ as well as that of IFNγ^+^IL-17A^+^ T cells (Figure 5B). Rg3 treatment slightly increased the frequency of IFNγ^+^IL-17A^−^ population; however, the overall frequency of IFNγ-producer was diminished due to a remarkable reduction in the frequency of IFNγ^+^IL-17A^+^ cells (vehicle vs. Rg3; 62.37 ± 1.90 vs. 45.70 ± 0.98, *p* = 0.0015). Thus, Rg3 hampered the reactivation and ex vivo expansion of MOG-reactive Th17 cells.

MOG-reactive Th17 cells are known to induce autoimmune neuroinflammation, and IL-17A has been proposed as a promising target for the treatment of multiple sclerosis (MS) [25]. Thus, we next tested whether Rg3 can ameliorate the severity of experimental autoimmune encephalomyelitis (EAE) in a mouse disease model for MS, induced by a transfer of MOG-reactive Th17 cells. Adoptive transfer of MOG-reactive ex vivo expanded Th17 cells (CD45.1^+/+^) is known to induce EAE (Figure 6A) [24]. As depicted in Figure 6B,C, recipients of Rg3-treated CD4^+^ T cells were more resistant to EAE induction concomitant with lesser weight loss, compared with the recipients of vehicle-treated CD4^+^ T cells. Accordingly, the clinical score was significantly lower in the former (Figure 6D), indicating that Rg3 treatment rendered MOG-specific autoreactive Th17 cells to be less pathogenic in inducing neuro-inflammation. Analysis of donor T cells harvested from the central nervous system (CNS) tissues showed that Rg3-treated CD4^+^ T cells exhibited a lower frequency of IL-17A^+^ population than vehicle-treated T cells (Figure 6E). Similarly, the IL-17A^+^ frequency among donor T cells was reduced in the draining inguinal lymph nodes (iLNs) of the former group (Figure 6F). In contrast, the frequency of IFN-γ^+^ cells among donor T cells in the CNS as well as in the iLNs was comparable between the two groups (Figure 6E,F). Together, these data suggest that Rg3 inhibits the reactivation of autoreactive Th17 cells rather than Th1 cells, leading to a diminished autoimmune neuro-inflammation associated with reduced Th17 cells in CNS tissues.

## 4. Discussion

While the immunomodulatory effects of KRG extract (KRGE) are well documented, the immuno-pharmacological effects of each component within the extracts are less clear. In this regard, ginsenoside Rg3 has been proposed to be immuno-modulatory based on the observation that Rg3-enhanced KRGE ameliorates various inflammatory diseases in animal models, including T-cell-mediated autoimmune diseases [2,4,10,18]. Among diverse helper T-cell subsets, Th17 cells appear to be critically pathogenic immune cells in autoimmune tissue inflammation, including psoriasis and multiple sclerosis. Therefore, in the present study, we aimed to investigate the role of Rg3 in Th17 cell differentiation and Th17-mediated autoimmune diseases by using animal models. Our in vitro studies revealed that Rg3 played only a minor role in the production of Th17-promoting cytokines IL-6, IL-12/23p40 and TNFα from DCs. Instead, Rg3 significantly inhibited the induction of RORγt expression in CD4^+^ T cells and therefore hampered the differentiation of Th17 cells from naïve precursors. Moreover, Rg3 appeared to inhibit the expansion of autoreactive MOG-specific Th17 cells during ex vivo restimulation with a cognate peptide, and thus also ameliorated the incidence and severity of EAE in the recipient mice. Thus, our findings suggest an anti-inflammatory mechanism by which Rg3 ameliorates autoimmune inflammation via inhibiting RORγt expression of CD4^+^ T cells during Th17 cell differentiation.

RORγt, a master transcription factor for Th17, is regulated by a complex regulatory circuit [26], but whether Rg3 intervenes with the RORγt expression has not been identified. Activation of STAT3 upon IL-6 signaling with concomitant TGFβ signaling induces RORγt expression [17]. Moreover, IL-23 further expands and stabilizes Th17 cells via STAT3. In this respect, it is noteworthy that Rg3 represses STAT3 phosphorylation in tumor cells by decreasing the hexokinase 2 level [27]. We observed that the transcript level of *Il21* in T cells stimulated with IL-6 and TGFβ was reduced by Rg3. IL-21 is known to be induced by IL-6 via STAT3 during Th17 cell differentiation [28]. Thus, it is feasible to hypothesize that Rg3 inhibits RORγt expression by suppressing STAT3 activation in T cells upon IL-6 and IL-23 signaling. Alternatively, it is possible that STAT5, a negative regulator of RORγt, was involved in the Th17 reduction by Rg3, given that ginseng can up-regulate STAT5 phosphorylation in cyclosporine-treated splenocytes poised to Treg differentiation [18]. It has also been delineated that natural ligands stemming from endogenous cholesterol metabolism can either promote or inhibit RORγt function by directly interacting with its binding sites [29]. Ursolic acid is one of the most well-characterized RORγt antagonists, which ameliorates EAE and allergic asthma [30,31]. Since Rg3 has a tetracyclic triterpenoid saponin structure [14], a homologue of the ursolic acid, Rg3 might belong to the specific RORγt antagonists as well. Further studies are required to dissect the direct molecular target(s) of Rg3 in developing Th17 cells and to elucidate the molecular mechanisms by which Rg3 regulates Th17 cell differentiation and expansion.

Several studies including our own showed that KRGE can modulate the production of innate cytokines from DCs and macrophages [10,11]. LPS stimulation induced the production of several Th17-promoting cytokines including IL-6, IL-23, TNFα, and IL-1β. The observed minor role of Rg3 in regulating the production of these cytokines from DCs suggests that other components within KRGE have DC-regulatory functions. The diminished RORγt expression by Rg3 treatment in T cells raises the possible role of Rg3 in regulating the functions of RORγt-expressing innate immune cells including type 3 innate lymphoid cells (ILC3) and LTi cells. Since these cells are known to play a crucial role in mucosal homeostasis and lymphoid tissue generation [32,33], it will be important to determine the potential role of Rg3 in these innate RORγt-expressing cells. Antibodies against IL-17A or IL-17RA are proven to be effective in the treatment of psoriasis in humans [34,35]. Thus, it will be of interest to investigate if Rg3 can ameliorate other autoimmune tissue inflammation, such as psoriasis.

In summary, the results of the present study indicate that Rg3 negatively regulates RORγt expression in CD4^+^ T cells, subsequently hindering Th17 cell differentiation and Th17-mediated neuro-inflammation. These findings are novel because they attest to the fact that ginsenoside Rg3 is the active constituent of KRG extract that can inhibit the differentiation of Th17 cells and Th17-cell-mediated autoimmunity. Our findings may provide a rationale for developing Rg3 as a potential candidate for the treatment of autoimmune disorders driven by Th17 cells.

## Figures and Tables

**Figure 1 biomolecules-10-00122-f001:**
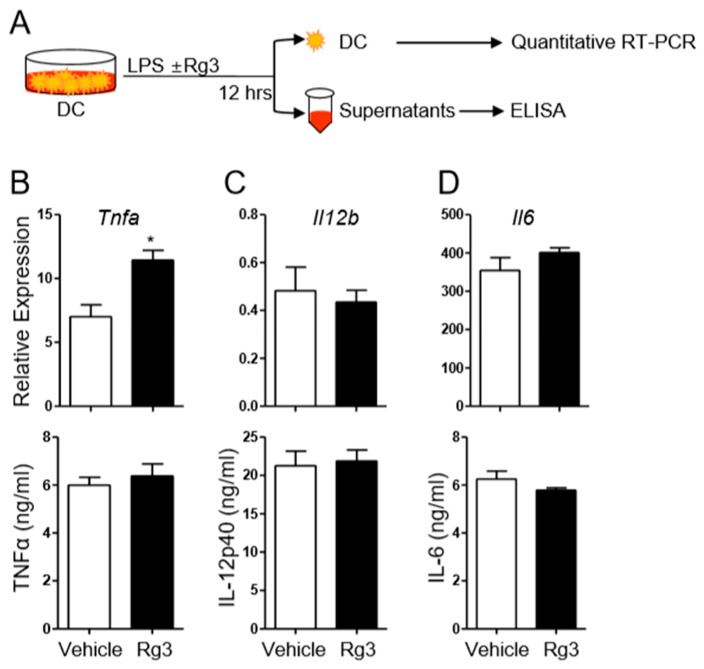
Effects of Rg3 treatment on the production of pro-inflammatory cytokines from LPS-stimulated bone-marrow-derived dendritic cells (BMDCs). (**A**) Experimental scheme. BMDCs were treated with LPS in the presence of DMSO (vehicle) or 37.5 μg/mL Rg3. After 12 h, the transcript and protein levels of indicated cytokines in BMDCs were measured by quantitative RT-PCR and ELISA (**B**–**D**). Data are mean ± SEM and represent three independent experiments. * *p* < 0.05.

**Figure 2 biomolecules-10-00122-f002:**
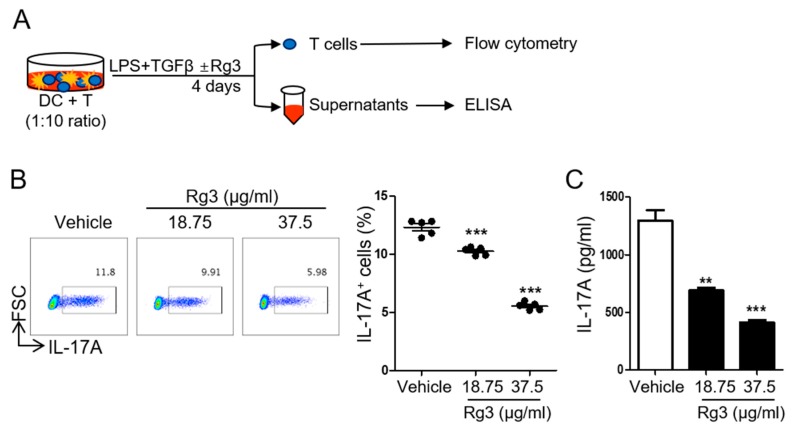
Rg3 inhibited DC-mediated Th17 differentiation. (**A**) Experimental scheme. Naïve CD4^+^ T cells were incubated with LPS-stimulated BMDCs in the presence of soluble anti-CD3 and 1.5 ng/mL TGFβ. DMSO (vehicle) or indicated doses of Rg3 (18.75 or 37.5 μg/mL) was added at the beginning of cell culture. The frequency of IL-17A^+^CD4^+^ T cells and the level of IL-17A were determined by FACS (**B**) and ELISA (**C**), respectively. Data are mean ± SEM and represent three independent experiments. ** *p* < 0.01, *** *p* < 0.001 in comparison with the vehicle-treated group.

**Figure 3 biomolecules-10-00122-f003:**
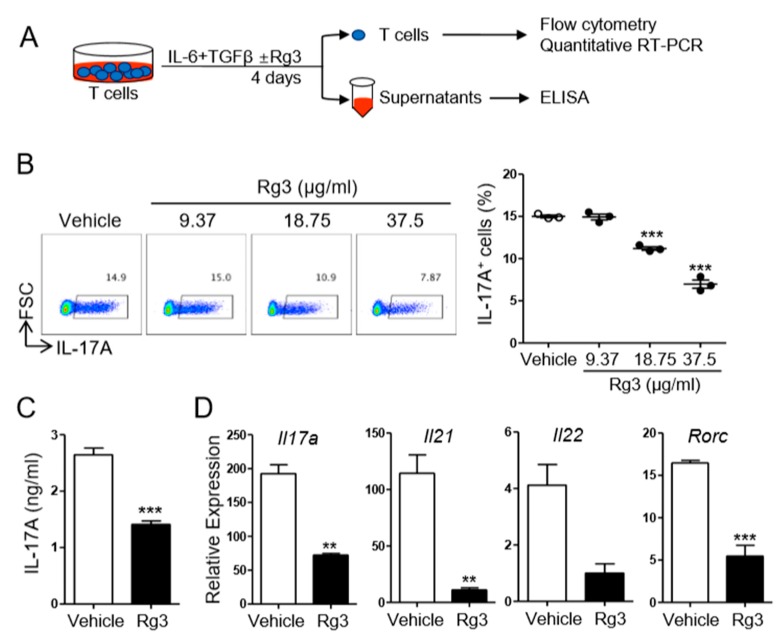
Rg3 inhibited Th17 differentiation in a T-cell-intrinsic manner. (**A**) Experimental scheme. Naïve CD4^+^ T cells were stimulated with pre-coated anti-CD3 and CD28 in the presence of 10 ng/mL IL-6 and 1.5 ng/mL TGFβ. DMSO (vehicle) or indicated doses of Rg3 (18.75 or 37.5 μg/mL) was added at the beginning cell culture. The frequency of IL-17A^+^CD4^+^ T cells and the level of IL-17A were determined by FACS. (**B**) The amount of IL-17A and the transcript levels of indicated genes were measured by ELISA and quantitative RT-PCR, respectively (**C**,**D**). The concentration of Rg3 in (**C**) and (**D**) was 37.5 μg/mL. Data represent mean ± SEM and represent three independent experiments. *** *p* < 0.001 in comparison with the vehicle-treated group.

**Figure 4 biomolecules-10-00122-f004:**
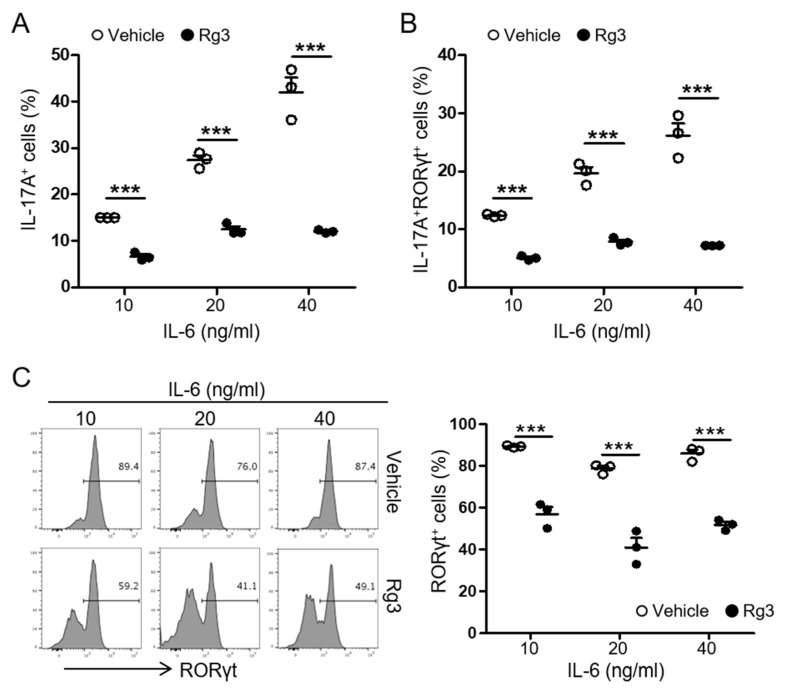
Rg3 diminished RORγt expression in T cells during Th17 differentiation. Naïve CD4^+^ T cells were stimulated with pre-coated anti-CD3 and CD28 in the presence of 1.5 ng/mL TGFβ and indicated concentrations of IL-6. DMSO (vehicle) or Rg3 (37.5 μg/mL) was added at the beginning of the cell culture. The frequency of IL-17A^+^ (**A**) and IL-17A^+^RORγt^+^ (**B**) CD4^+^ T cells was measured by FACS. (**C**) The histogram of RORγt^+^ expression in CD4^+^ T cells is shown. Experiments were conducted three times. Data represent mean ± SEM. *** *p* < 0.001 in comparison with the vehicle-treated group.

**Figure 5 biomolecules-10-00122-f005:**
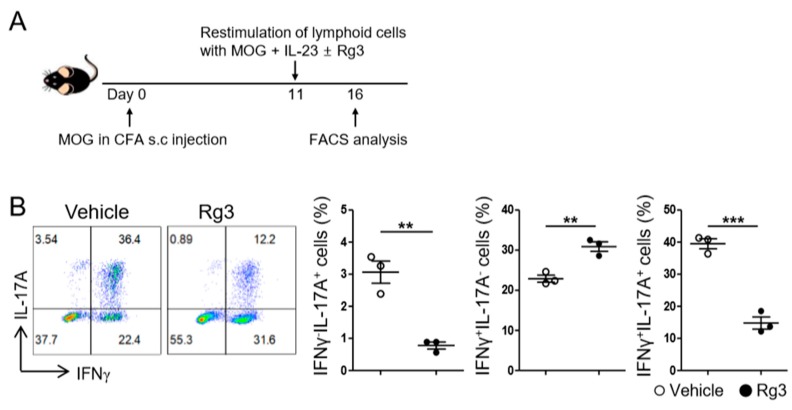
Rg3 inhibited the expansion of myelin oligodendrocyte glycoprotein (MOG)-reactive Th17 cells. (**A**) Experimental scheme. Mice were immunized s.c. with MOG in CFA. Seven days later, the lymphoid cells from the draining lymph nodes (LNs) were harvested and restimulated with MOG and IL-23 in the presence of DMSO (vehicle) or Rg3 (37.5 μg/mL) for 5 days. (**B**) The expression of IL-17A and IFNγ in CD4^+^ T cells were examined by FACS. Data shown are mean ± SEM and represent one from two independent experiments. ** *p* < 0.01, *** *p* < 0.001 in comparison with the vehicle-treated group.

**Figure 6 biomolecules-10-00122-f006:**
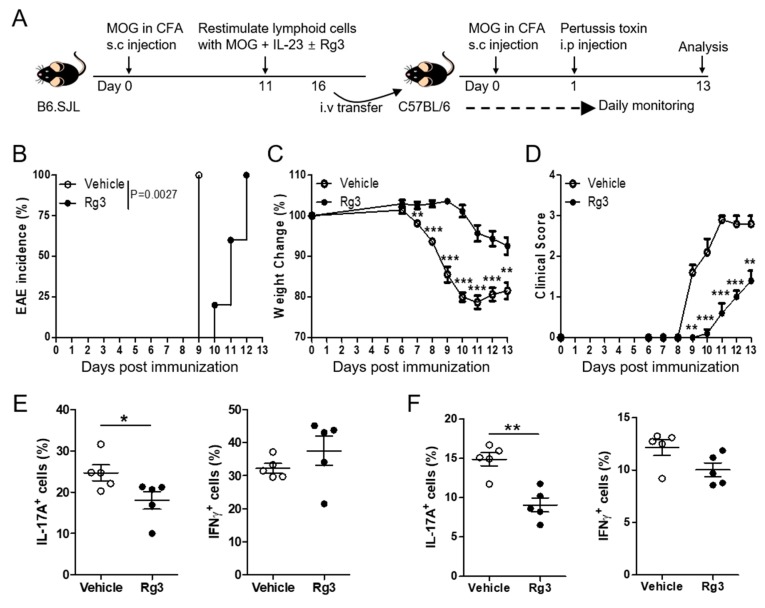
Rg3 decreases autoimmune neuro-inflammation induced by adoptive transfer of MOG-reactive CD4^+^ T cells. (**A**) Experimental scheme of passive experimental autoimmune encephalomyelitis (EAE), induction. (**B**–**D**) EAE incidence, % weight change, and clinical score were monitored daily. The levels of IL-17A and IFNγ in CD4^+^ T cells were examined in the CNS (**E**) and the inguinal LN (**F**) on day 13 after EAE induction. Data shown are mean ± SEM and represent two independent experiments. * *p* < 0.05, ** *p* < 0.01, *** *p* < 0.001 in comparison with the vehicle-treated group.

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
