# Peer review of "A Critical Regulation of Th17 Cell Responses and Autoimmune Neuro-Inflammation by Ginsenoside Rg3"

_biomolecules, 2020, doi:10.3390/biom10010122_

Round 1

Reviewer 1 Report

The study provided the role of Rg3 on Th17 cells and Th17-mediated autoimmunity. The content of the paper is quite valuable, but there are some problem in experimental design should be solved before the manuscript been considered for publication.

Q1: Fig 1B compares inflammatory factor with and without adding Rg3 after detecting DC stimulation by LPS. Please explain why TNF-α expression in the Rg 3 group is significantly higher than the mRNA level of supernatants protein?

Q2: In Fig 4C, the results of Rg3 groups are not consistent with the statistics on the right. Please correct them.

the IL-6 expressions on the left in 40ng / ml,

Q3: In Fig 5B, Rg3 groups is not consistent with the statistics on the right. Please correct the expressions of IFNγ + by flow cytometry.

Q4: In page 227-231, the result of of IFNγ + in Fig. 6 E & F was not described in the Results, Please add it !

Q5: In Figure 5, The description of Fig. 6E is similar to the result of 6F that was inconsistent with the IFNγ + result, please correct it !

Q6: The animal model using MOG with CFA (adjuvant) to increase immune stimulation to establish an autoimmune neuro-inflammation model. Does the experiment have a CFA (adjuvant) group as a control group? There are other papers shows that CFA (adjuvant) itself can induce rheumatoid Arthritis (Ref. 1 & 2). Please discuss the relationship between CFA with heat inactivated M. tuberculosis and the EAE model established in this experiment.

Reference

Omnia Ahmed Mohamed Abdel El- Gaphara, Amira Mourad Abo-Youssefb* and

Ali Ahmed Abo-Saif. Effect of Losartan in Complete Freund’s Adjuvant –Induced Arthritis in Rats. Iranian Journal of Pharmaceutical Research (2018), 17 (4): 1420-1430

Qi Xu,1 Yong Zhou,2 Rong Zhang,3 Zhan Sun,3 and Lu-feng Cheng. Antiarthritic Activity of Qi-Wu Rheumatism Granule a Chinese Herbal Compound) on Complete Freund’s Adjuvant-Induced Arthritis in Rats. Evidence-Based Complementary and Alternative Medicine. Volume( 2017), Article ID 1960517, 13 pages. https://doi.org/10.1155/2017/1960517

Author Response

Point-by-point responses

We appreciate the chance to respond to the reviewers’ comments with a revised manuscript. We have carefully considered all issues raised by the reviewers and submit a revised manuscript. As suggested by Reviewer #2, we also had a native speaker, Mr. Chris Daehong Kim, to correct grammatical errors throughout the manuscript. For clarity, we have underlined all changes in the revised manuscript. Below is our point-by-point response to the reviewers’ comments.

Reviewer #1:

Q1: Fig 1B compares inflammatory factor with and without adding Rg3 after detecting DC stimulation by LPS. Please explain why TNF-α expression in the Rg 3 group is significantly higher than the mRNA level of supernatants protein?

RE: We appreciate the reviewer’s careful comment. Although there is a positive correlation between the level of transcript and protein in most cases, an increase in the level of a transcript of a certain gene does not necessarily mean an increase in the level of a protein encoded by the gene. The discrepancy between transcript and protein could be due to multiple processes beyond transcription as recently reviewed (Y. Liu et al., Cell, 2016; 165(3):535-50). To demonstrate this point, we have revised the Results section (page 8).

Q2: In Fig 4C, the results of Rg3 groups are not consistent with the statistics on the right. Please correct them.

RE: We thank the reviewer’s important point-out. We have mistakenly incorporated an incorrect histogram in the IL-6 (40 ng/ml) plus Rg3-treated T cells in Figure 4C. We have exchanged it with the correct histogram in the revised manuscript. We regret this mistake and appreciate the reviewer’s correction with this matter.

Q3: In Fig 5B, Rg3 groups is not consistent with the statistics on the right. Please correct the expressions of IFNγ + by flow cytometry.

RE: The frequency of the total IFNγ+ T cells was reduced by Rg3 treatment while that of IFNγ+IL-17A- T cells was slightly increased. To avoid this confusion, we have reanalyzed the data to separately describe IFNγ+IL-17A-, IFNγ+IL-17A+, IFNγ+IL-17A+ populations in Fig. 5B in the revised manuscript. We have revised the Results section accordingly to reflect this new analysis (page 10-11).

Q4: In page 227-231, the result of of IFNγ + in Fig. 6 E & F was not described in the Results, Please add it !

RE: As the reviewer suggested, we have revised the Results section to describe the results of IFNγ+ cells more clearly in the revised manuscript (page 11).

Q5: In Figure 5, The description of Fig. 6E is similar to the result of 6F that was inconsistent with the IFNγ + result, please correct it!

RE: In Figure 6E & F, we have analyzed the donor T cells in the recipient mice of ex vivo-expanded MOG-reactive T cells after additional immunization with MOG in CFA in Fig. 6E & F. While we observed a decrease in the frequency of IFNγ+ cells after Rg3 as described in Fig. 5B (donor T cells before transfer), we observed comparable frequencies of IFNγ+ donor T cells both in the CNS and in the iLN in Fig. 6E & F (donor T cells after transfer and additional immunization). We do not think that these results are conflicting since they were from different experimental systems (with or without additional immunization). Th17 cells are known to be plastic and can be further transdifferentiated into Th1 cells upon restimulation (Lee YK et al. Immunity. 2009. 30:92-107). Our results suggest that the suppression of Th17 cell responses by Rg3 was maintained even after restimulation with the cognate antigen in vivo, while that of Th1 cell responses failed to do so. To describe this point more clearly, we have revised a sentence in the results section (page 11).

Q6: The animal model using MOG with CFA (adjuvant) to increase immune stimulation to establish an autoimmune neuro-inflammation model. Does the experiment have a CFA (adjuvant) group as a control group? There are other papers shows that CFA (adjuvant) itself can induce rheumatoid Arthritis (Ref. 1 & 2). Please discuss the relationship between CFA with heat inactivated M. tuberculosis and the EAE model established in this experiment.

Reference

Omnia Ahmed Mohamed Abdel El- Gaphara, Amira Mourad Abo-Youssefb* and Ali Ahmed Abo-Saif. Effect of Losartan in Complete Freund’s Adjuvant –Induced Arthritis in Rats. Iranian Journal of Pharmaceutical Research (2018), 17 (4): 1420-1430

Qi Xu,1 Yong Zhou,2 Rong Zhang,3 Zhan Sun,3 and Lu-feng Cheng. Antiarthritic Activity of Qi-Wu Rheumatism Granule a Chinese Herbal Compound) on Complete Freund’s Adjuvant-Induced Arthritis in Rats. Evidence-Based Complementary and Alternative Medicine. Volume( 2017), Article ID 1960517, 13 pages.

RE: We appreciate the reviewer’s comment on the issue of control. For the experiment described in Figure 6, we have aimed to investigate the effect of Rg3 on autoreactive CD4 T cells in an adoptive transfer EAE model. We have used vehicle treatment as a control in our experiment. All the recipients were immunized with MOG in CFA. As the reviewer suggested, the addition of CFA (without MOG emulsion) would further strengthen the finding. At the same time, we believe that comparing vehicle-treated vs Rg3-treated is sufficient to draw the conclusion that Rg3 treatment impacts the expansion of autoreactive T cells.  Moreover, unlike rat arthritis model induced by paw pad injection of CFA, EAE induction exclusively depends on immunized encephalitogenic peptides (Robinson AP et al., Handb Clin Neurol., 2014; 122:173-189). Thus, CFA-treated group may not be necessary in the setting of the MOG-induced EAE model in our experimental setting.

Reviewer 2 Report

The manuscript submitted by Park etl is about ginsenoside Rg3 alleviate Th17 cell differentiation and Th17 mediated neuro-inflammation. Firstly, it was found out that Rg3 suppress Th17 cell differentiation via inhibiting the expression of RORγt in CD4+ T cell during in vitro, and administration of Rg3 in MOG35−55-induced chronic EAE mice reduced the reactivation of MOG-activated Th17 cells.

Several suggestions is shown as follows:

1. One significant challenge in this field is the Novelty of study, there is plenty of studies reporting the anti-inflammatory effects of ginseng and ginsenosides on Th17 cells (see for example Lee et al., Mol Neurobiol (2016) 53:1977–2002). The authors should let us know what's the novelty behind and why it is still necessary to further study.

In cell experiment, it has been found that Rg3 modulation in Th17 cell is associated with RORγt in CD4+ T cells, however, the interference of Rg3 in animal experiment didn’t mention that the inactivation of Th17 cells is associated with RORγt or not. The evidence of animal experiment that support the Rg3 induced Th17 regulation should be provided.

This article indicated that Rg3 administration could evoke the reduction of Th17 cell differentiation and RORγt expression via flow analysis, however, it didn’t mentioned that how Rg3 triggered Th17 cell differentiation interactive pathway, and the mRNA and protein expression level of cytokines involved in the Th17-RORγt pathway should be detected.

Above all, The content of this article didn’t provide a comprehensive response of Th17 cell after Rg3 treatment.

Author Response

Point-by-point responses

We appreciate the chance to respond to the reviewers’ comments with a revised manuscript. We have carefully considered all issues raised by the reviewers and submit a revised manuscript. As suggested by Reviewer #2, we also had a native speaker, Mr. Chris Daehong Kim, to correct grammatical errors throughout the manuscript. For clarity, we have underlined all changes in the revised manuscript. Below is our point-by-point response to the reviewers’ comments.

Reviewer #2:

Q1: One significant challenge in this field is the Novelty of study, there is plenty of studies reporting the anti-inflammatory effects of ginseng and ginsenosides on Th17 cells (see for example Lee et al., Mol Neurobiol (2016) 53:1977–2002). The authors should let us know what's the novelty behind and why it is still necessary to further study.

 RE: We appreciate the reviewer for raising this important point. Most of the published works related to ginseng and Th17 responses have been focused on the role of ginseng extracts in Th17 cell responses. For instance, it has been reported that systemic administration of Korean red ginseng extracts renders mice more resistant to EAE induction with decreased Th17 cells. On the other hand, only a few studies have addressed the role of individual ginsenoside, particularly Rg3, in regulating autoimmune T cell responses. Since several studies have shown the regulatory effect of red ginseng extracts on Th17 cell-mediated diseases (Kim KR et al., Biol Pharm Bull. 2010;33(4):604-10, Jhun J et al., Mediators Inflamm. 2014; 2014:351856, Lee MJ et al., Mol Neurobiol. 2016;53(3):1977-2002), it is important to determine which constituent in the ginseng extract could regulate the differentiation and function of Th17 cells. We believe that our study provides novel findings since we demonstrate that ginsenoside Rg3 can inhibit Th17 cell differentiation and thus can ameliorate autoimmune neuroinflammation mediated by autoreactive Th17 cells. Therefore, we propose that Rg3 is one of the major constituents in ginseng extracts that can inhibit Th17 cell responses. We have added sentences in the Discussion section to clarify the novelty of the present study (page 14).

Q2. In cell experiment, it has been found that Rg3 modulation in Th17 cell is associated with RORγt in CD4+ T cells, however, the interference of Rg3 in animal experiment didn’t mention that the inactivation of Th17 cells is associated with RORγt or not. The evidence of animal experiment that support the Rg3 induced Th17 regulation should be provided.

RE: We appreciate the reviewer’s comment. We agree that the analysis of RORγt in T cells from an animal experiment can further support our main conclusion. We regret that we were not able to include a new set of data that can address this point due to time limitation since we were asked to submit the revision within 10 days by the editorial office. Nevertheless, the results in Figure 6 clearly demonstrate that the amelioration of EAE in the Rg3-treated group is associated with decreased Th17 cell responses but not with Th1 cell responses. Thus, we believe that our findings support the notion that Rg3 treatment can diminish Th17 cell responses in the recipient mice.

Q3. This article indicated that Rg3 administration could evoke the reduction of Th17 cell differentiation and RORγt expression via flow analysis, however, it didn’t mention that how Rg3 triggered Th17 cell differentiation interactive pathway, and the mRNA and protein expression level of cytokines involved in the Th17-RORγt pathway should be detected.

RE: We thank the reviewer for this constructive suggestion. We have further analyzed if Rg3 can impact the expression of Th17-associated factors including Il17a, Il21, Il22, and Rorc by quantitative RT-PCR, and have added the results as Figure 3D in the revised manuscript. Our new analysis showed that the addition of Rg3 remarkably reduced the expression of these Th17-associated transcript, further supporting the negative regulation of Th17 cell differentiation by Rg3 treatment. We have added sentences to describe these new data in Results (page 9) and in Discussion sections (page 12-13).

Round 2

Reviewer 1 Report

The authors have responded and modified most of the suggestions in the revised manuscript. If the editor agrees that the content of the revised manuscript meets the quality and purpose of the journal, it can be considered for publication.

Reviewer 2 Report

Nil